# Thin-Film Composite Nanofiltration Membranes for Non-Polar Solvents

**DOI:** 10.3390/membranes11030184

**Published:** 2021-03-09

**Authors:** Seungmin Lee, Taewon Kang, Jong Young Lee, Jiyu Park, Seoung Ho Choi, Jin-Yeong Yu, Serin Ok, Sang-Hee Park

**Affiliations:** 1Energy Materials and Components R&D Group, Korea Institute of Industrial Technology, Busan 46938, Korea; leesm0506@kitech.re.kr; 2Department of Chemical Engineering, Changwon National University (CNU), Changwon 51140, Korea; tae6467@gs.cwnu.ac.kr (T.K.); pule4546@gs.cwnu.ac.kr (J.Y.L.); greenair6@gs.cwnu.ac.kr (J.P.); tmdgh234@gs.cwnu.ac.kr (S.H.C.); oijy0102@gs.cwnu.ac.kr (J.-Y.Y.); sl0139@gs.cwnu.ac.kr (S.O.)

**Keywords:** thin-film composite membranes, organic solvent nanofiltration membrane, selective layer, interfacial polymerization, solvent resistance

## Abstract

Organic solvent nanofiltration (OSN) has been recognized as an eco-friendly separation system owing to its excellent cost and energy saving efficiency, easy scale-up in the narrow area and mild operation conditions. Membrane properties are the key part in terms of determining the separation efficiency in the OSN system. In this review paper, the recently reported OSN thin-film composite (TFC) membranes were investigated to understand insight of membrane materials and performance. Especially, we highlighted the representative study concepts and materials of the selective layer of OSN TFC membranes for non-polar solvents. The proper choice of monomers and additives for the selective layer forms much more interconnected voids and the enhanced microporosity, which can improve membrane performance of the OSN TFC membrane with reducing the transport resistance. Therefore, this review paper could be an important bridge to connect with the next-generation OSN TFC membranes for non-polar solvents.

## 1. Introduction

Organic solvent nanofiltration (OSN) is a pressure-driven separation process, applicable for the solvent recycle and exchange, concentrations, and purification of chemicals and pharmaceuticals in organic solvent environments, which is also called as solvent resistant nanofiltration (SRNF) [1,2]. In general, OSN is an emerging platform to effectively separate solutes with molecular weights ranging from 100 to 1000 g mol^−1^ from polar or non-polar solvent solution, which could replace the conventional separation process such as the adsorption, extraction, distillation, evaporation, and chromatography [3]. 

With respect to the sustainability, the membrane-based OSN separation process has been recognized as a best separation technology due to its low energy requirements, low solid waste generation, low carbon footprint, low labor intensity, straightforward scale-up, stability in harsh environments (pH, high temperature, and solvents), mild operating conditions (room temperature and low pressure), easy solvent swap from high to low boiling point solvents, and simultaneous removal of solutes from different chemical environments [4,5].

For last a few decades, polymeric membranes have been intensively explored to apply for diverse OSN separation processes due to their huge merits including the large variety of available polymers, relatively low price, simple fabrication process, and ease of upscaling [6]. Two kinds of main polymeric membranes are integrally skinned asymmetric (ISA) and thin-film composite (TFC) membranes [7]. ISA membranes are usually prepared by the phase inversion technique, resulting in the formation of dense selective layer with a few hundred nanometers in thickness on highly porous sublayer with several microns in thickness [8]. Despite of their advantages including easy fabrication process and high mechanical strength, the permeance reduction by its physical aging and compaction over time in operation is a critical issue to be solved [9]. 

Thin-film composite (TFC) membranes are commonly prepared by diverse coating methods, which are composed of an ultra-thin selective layer on top of an ISA-type porous support membrane [10]. Compared with ISA membrane, the selective layer was easily controlled, leading to better separation performance. In this review paper, it is mainly described which polymeric materials have been used for the selective layer and support membrane of TFC membranes. Especially, the objective of this review is to announce the advances in TFC membranes prepared with different materials for the selective layer, applicable to non-polar solvents such as toluene, *n*-hexane, and *n*-heptane. In addition, this study may provide useful insights and directions for the development of the next-generation OSN membranes.

## 2. Materials for TFC Membrane

TFC membranes consists of an ultra-thin selective layer on top of a highly porous asymmetric support membrane. The selective layer can be usually formed via the interfacial polymerization (IP) of two monomers of coating polymers or prepolymers. In the OSN applications, the selective layer mainly determines the solvent permeance and selectivity between solvent and solutes, the support membrane should have an excellent chemical stability against diverse organic solvents and the high mechanical and thermal stabilities. Thus, many studies have focused on the design for the selective layer and the support membrane independently to obtain the optimum performance. In this section, the recently reported organic monomers and polymers to prepare TFC OSN membranes are described.

### 2.1. Support Membranes

The asymmetric support membranes are commonly fabricated by a non-solvent induced phase inversion separation (NIPS) technique that includes three steps as follows: Preparation of the polymer dope solution using polar aprotic solvents such as dimethylformamide (DMF), *N*-methyl-2-pyrrolidone (NMP), and dimethyl sulfoxide (DMSO) and degassingCasting the polymer film on the highly porous non-woven fabric using a casting knifeImmersing the cast film in the non-solvent (usually water) bath

When the cast film is immersed in non-solvent bath, the solidification of the polymer immediately occurs through the solvent exchange between the polar aprotic solvent and non-solvent, leading to the formation of the asymmetric porous support membrane. This is the simple and most powerful technique to generally prepare the ultrafiltration (UF)-grade and nanofiltration (NF)-grade asymmetric membranes. It has been usually used in water purification applications. It is also called as an ISA membrane in OSN applications. During the phase inversion separation step, the dense skin layer is rapidly formed by the fast rate of solvent exchanging, while the porous sublayer with large pores is slowly formed due to the relatively slow rate of solvent exchanging. Therefore, the asymmetric porous membrane is formed, which is composed of a thin and dense skin layer and a thick and more porous sublayer.

It is a key point to control the thermodynamic stability of the phase inversion system and the speed of solvent exchanging, in terms of the formation of membrane morphology (i.e., asymmetric structures, thickness, and porosity of the skin layer) and the membrane performance. In 2011, Livingston research group demonstrated the main membrane formation parameters, including polymer/solvent/non-solvent system, evaporation step, and the role of a co-solvent, and influence of polymer characteristics on membrane performance during the phase inversion process [11,12,13].

To prepare the porous support membrane, various materials have been commonly used such as polysulfone (PSF) [14,15,16,17,18,19], polyethersulfone (PES) [20,21], poly(ether ether ketone) (PEEK) [22,23,24], polyacrylonitrile (PAN) [25,26], poly(vinylidene fluoride) (PVDF) [27,28,29], polyethylene (PE) [30], polypropylene (PP) [31,32,33], the recycled polyethylene terephthalate (PET) [34], and cellulose [35] as shown in Table 1. Most of the porous support membranes are prepared by the phase inversion method, while the common polyolefin membranes including PE and PP are manufactured by sequential steps consisting of the melt extrusion and mechanical stretching [36]. Porous PP and PE membranes are the strong candidate as a support membrane for OSN applications owing to their great mechanical and thermal properties and excellent resistance against most common polar and non-polar solvents. However, the common polymeric membranes such as PSF and PES shows the poor solvent resistance for strong polar solvents including DMF and DMSO etc., since it is fabricated from the polymeric dope solution of the same strong polar solvents, obstructing the expansion of their OSN application areas without a further cross-linking step. 

Here, there are the representative cross-linked porous supports that are prepared through the reaction with multi-functional cross-linking agent after forming the porous structures by the phase inversion separation technique as shown in Figure 1. One is a cross-linked polybenzimidazole (PBI) membrane, which can be prepared via the cross-linking reaction with multi-functional alkyl halides [37,38,39], epoxies [40], and acyl chlorides [41]. These reactions lead to the formation of covalent carbon-nitrogen bonds, resulting in showing the excellent solvent resistance of the porous support membrane. Other cases are cross-linked P84 and Materimid polyimide (PI) membranes. These polymers can form amide bonds through the cross-linking reaction with multi-functional amines [42,43]. To date, the cross-linking approach is the best method to prepare the OSN membrane with the great solvent resistance, while these materials are so expensive and brittle [36]. In particular, the cross-linking step of PI support membranes is time consuming, which takes more than 16 h, contributing to the enhancement of manufacturing cost and the production of chemical wastes. 

Recently, some different cross-linking methods have been reported to obtain solvent-stable porous supports. For example, Zhao et al. [44] reported the interpenetrating polymer networking (IPN) concept using PBI and polydopamine (PDA). PDA polymerization is conducted by immersing in the NIO_4_ and Tris buffer solution for several days after the phase inversion step of the PBI/PDA dope solution. The prepared porous membrane showed excellent thermal stability and solvent resistance. In addition, Lu et al. [45] and Kim et al. [46] have prepared polyamic acid (PAA) and hydroxy polyimide (HPI) precursor polymers, respectively. Both PI and HPI nanofiber mats are prepared by an electrospinning of each solution, which revealed great solvent resistance after the heat-treatment at 300~400 °C.

### 2.2. Selective Layers

As mentioned above, a selective layer of TFC membranes determines the final membrane performance, which can be fabricated by interfacial polymerization [47], coating [8,48], 3D printing [49,50], and self-assembly layer-by-layer [51,52] techniques on the surface of a porous support membrane. In general, polyamide (PA)-based selective layer is prepared via interfacial polymerization of *m*-phenylenediamine (MPD) and trimesoyl chloride (TMC), where two monomers rapidly react at the interface between two immiscible water and organic solvent phases [53]. The three-dimensional (3D) cross-linked PA-selective layer is formed by the reaction combination between two amine groups of MPD and three acid chloride groups of TMC. In addition, the dense and highly cross-linked PA-selective layer is formed in a few seconds due to the high reactivity of acyl chlorides with amines. The whole interfacial polymerization process contains multisteps, but it is so easy and short as follows:Preparation of two immiscible monomer solutions (MPD and TMC commonly dissolved in water and *n*-hexane, respectively)First step: immersing the porous support membrane in MPD aqueous solution for a few minutes (3~30 min)Second step: contacting MPD-immersed support membrane in TMC organic solution for a few minutes (1~5 min) after removing excess MPD solution on the porous support membrane using a gummous roller.Final step: cleaning the prepared PA TFC membrane using a pure *n*-hexane to stop the reaction and then drying.

Thus, the simple and cost-effective interfacial polymerization technique has been widely used for fabricating TFC membranes in both the academic world and industrial applications like as nanofiltration (NF) and reverse osmosis (RO) process. 

#### 2.2.1. Conventional MPD-based Selective Layer

Since Solomon et al. have first introduced PA TFC membrane as a OSN membrane, many researchers have reported PA TFC membranes prepared using multi-functional amines and acid chlorides on various types of support membranes [10]. They prepared the selective layer using MPD/piperazine (PIP) and TMC on the cross-linked PI support membrane to obtain the stable membrane against DMF. Compared to the conventional ISA OSN membranes (first generation membrane), the developed TFC membranes showed excellent solvent permeance without sacrificing selectivity, which has been considered as a strong candidate of the second generation OSN membrane.

In 2015, the ultra-thin selective layer with only 8 nm thickness was successfully prepared via interfacial polymerization of MPD and TMC as shown in Figure 2 [54]. PIP and 4-(aminomethyl)piperidine (AMP) were also used as a monomer in an aqueous phase instead of MPD, leading to the formation of MPD-based and/or PIP-based polyamide selective layer. The ultra-thin polyamide selective layer with the flat surface morphology was fabricated by introducing the highly porous cadmium hydroxide (Cd(OH)^2^) nanostrand as a middle layer on top of porous cross-linked PI or alumina. Performance of the developed TFC membranes was systemically investigated with different MPD and TMC concentrations, reaction times, and types of amine monomers. It is interesting to note that the significantly enhanced permeance of the MPD-based selective layer was observed after DMF activation without the noticeable decline of rejection but was not revealed at the selective layers prepared with other monomers (PIP or 4-(aminomethyl)piperidine (AMP)) [54]. 

Recently, it was experimentally demonstrated that unreacted monomers and small PA fragments are extracted from PA network during the solvent activation [55]. Therefore, it can be reasonably postulated that the high permeable pathways inside PA networks are formed by removing the small PA fragments. Nevertheless, this solvent activation effect of a PA-selective layer should be carefully studied, because its chemical structures and swelling effects can be dramatically changed by the monomer kinds and concentrations.

For the MPD-based PA-selective layer, it should be noted that the use of a surfactant dramatically increases in membrane performance even if it also increases the thickness and thus transport resistance of the selective layer. Park et al. have reported the TFC OSN membrane composed of a very thick selective layer on top of PE porous support membrane using sodium dodecyl sulfate (SDS) as shown in Figure 3 [56]. To successfully prepare the MPD-based selective layer on the hydrophobic PE support membrane (contact angle = 120°), the PE support membrane was O_2_ plasma-treated to provide the hydrophilic nature. In addition, the surfactant (SDS) as an additive was added in a MPD aqueous solution, resulting in the improvement of the wettability of the support with a MPD aqueous solution and the formation of the stable reaction interface between a MPD aqueous solution and a TMC organic solution.

Compared to the pristine selective layer without SDS, the thickness and roughness of the selective layer increased more than 6 and 2.5 times, respectively. In general, the thick selective layer has the high transport resistance, leading to the decline of the solvent permeance [56,57]. Nevertheless, acetone permeance of the TFC membrane prepared by the assistance of 0.2 wt% SDS was significantly enhanced more than 6 times and the styrene oligomer rejection dramatically increased. The enhanced acetone permeance is ascribed to the increased surface roughness that overwhelms the increased thickness. In addition, the observed high rejection indicates the formation of the highly cross-linked PA-selective layer. 

Thus, the use of the proper amount of a surfactant and/or some additives could help to the formation of the highly cross-linked and more permeable PA-selective layer. In terms of solvent permeance, it should be mentioned that the developed TFC OSN membranes prepared by MPD and TMC showed great permeance enhancement in polar solvents, such as acetonitrile, acetone, and methanol, but not in non-polar solvents, such as toluene, *n*-hexane, and *n*-heptane. This is because, these TMC-based TFC membranes have the relatively hydrophilic selective layer (contact angle = 60~70°) due to a lot of carboxylic acid groups, which are donated by the hydrolysis of TMC [58,59]. Therefore, new design of a selective layer is necessary to obtain high permeable TFC membrane for non-polar solvents. 

#### 2.2.2. Selective Layer with Enhanced Microporosity 

Here, some highly permeable and selective membranes for non-polar solvents have been introduced. Among them, Solomon et al. [60] have reported the defect-free and high cross-linked polyester selective layer prepared by interfacial polymerization of TMC and diverse contorted phenols, instead of MPD on top of a cross-linked PI or an alumina support as shown in Figure 4. It was their hypothesis that the contorted monomers can provide non-coplanar orientations in the film networks, increasing interconnectivity of intermolecular voids, which are beneficial for enhancing membrane performance. 

To demonstrate the hypothesis, diverse phenols, which are contorted monomers of spiro-structured 5,5′,6,6′-tetrahydroxy-3,3,3′,3′-tetra- methylspirobisindane (TTSBI) and cardo-structured 9,9-bis(4-hydroxyphenyl) fluorene (BHPF) and non-contorted monomers of dihydroxyanthraquinone (DHAQ), and 1,3-benzenediol (RES), were used to react with TMC. Compared to amine groups, the relatively slow reactivity of hydroxyl groups donated from a phenol was greatly promoted by adding sodium hydroxide (NaOH) with molar ratio of 4:1 (NaOH to monomer). The prepared BHPF- and TTSBI-based polyester contorted films showed much higher solvent permeance in both polar and non-polar solvents than that of the DHAQ- and RES-based polyester non-contorted films. Indeed, much more interconnected voids and the enhanced microporosity were formed by the contorted structures of BHPF and TTSBI, which was demonstrated through the three-dimensional (3D) molecular modeling study. 

Furthermore, polymers of intrinsic microporosity (PIMs) have been explored to fabricate the high permeable OSN membranes. It has a great attractive merit as a selective layer with the high free volume donated from its continuous network of interconnected intermolecular voids like the polyester contorted film mentioned above [8]. PIM-based selective layers have been usually prepared on porous support membranes by dip coating [61,62], roll-to-roll dip coating [63,64], or spin coating [8,48]. 

Gorgojo et al. [8] have reported that ultra-thin PIM film with intrinsic microporosity and 35 nm in thickness is prepared on a glass substrate by a spin coating method, subsequently transferred to a porous PAN or an alumina support as shown in Figure 5. Thickness of the PIM-selective layer was systemically controlled by the PIM concentrations dissolved in chloroform (CF). The prepared PIM-based TFC membrane showed excellent non-polar solvent (*n*-heptane) permeance with a rejection for 90% of hexaphenylbenzene.

Finally, two-dimensional (2D) covalent organic frameworks (COFs) are one of the promising candidates as a selective layer due to their crystalline polymeric networks with well-defined and inherent porosities [65]. The ordered and nano-sized pore channels (0.7~4.7 nm) of COF thin films are beneficial for enhancing solvent permeance. In addition, the chemical structures with the covalent bonds between six elements of H, B, C, N, O, and Si provide the great solvent and thermal resistance, low density, and high surface area, which are suitable for OSN applications. According to the type of monomers and reactions, COF thin films are commonly classified as three kinds of boron-, imine-, and triazine-based COF as shown in Figure 6 [66,67]. 

The boron-based COF thin film is synthesized by the self-condensation of boronic esters or the condensation reaction of boronic acid and catechol groups. It is a good candidate for the selective layer for OSN application, but the hydrolysis property of its ester-based chemical structures is the significant weak point to use as a selective layer, compared to other COF thin films. On the other hand, imine-based COF thin film is prepared by the condensation reaction between amine and aldehyde groups, which is less susceptible to the hydrolysis than boron-based one. The outstanding solvent resistance against diverse organic solvents has been demonstrated by other pioneer researchers [68,69,70]. Additionally, the triazine-based COF also showed lower crystallinity and higher stability than boron-based COF, which is stable in diverse organic solvents [71].

COF-based membranes have been fabricated by the solution casting, assembly of COF nanosheets, solvothermal synthesis, mechanochemical synthesis, and interfacial polymerization techniques [72,73]. Matsumoto et al. reported crystalline and the free-standing COF thin film prepared through the interfacial polymerization of polyfunctional amines and aldehydes [74]. They used scandium(ΙΙΙ) triflate (Sc(OTf)_3_) as a Lewis-acid catalyst to promote imine reaction, resulting in the formation of the continuous imine-based COF thin film. The film thickness was controlled by the monomer concentrations. Especially, the ultra-thin film with 2.5 nm thickness was successfully fabricated at low monomer concentrations. The results are most valuable in terms of that the interfacial polymerization system using Sc(OTf)_3_ catalyst would help to easily fabricate new class of COF-based OSN TFC membranes. 

In addition, Liang et al. [75] recently fabricated the conjugated microporous polymers (CMP) membrane with about 40 nm thickness via surface-initiated Sonogashira–Hagihara polymerization between 1,3,5-triethylnylbenzene and dibromobenzenes on the surface of the bromobenzene-functionalized silicon wafer (Si/SiO_2_ substrate) as shown in Figure 7. It was transferred to the top of porous PAN support to test membrane performance. As a result, the prepared CMP membrane showed ultra-fast *n*-hexane permeance (32 L m^−2^ h^−1^ bar^−1^) including dye rejection with molecule-weight cutoff (MWCO) of ~560 g mol^−1^. Therefore, above two cases could be the excellent evidence to provide new opportunity to fabricate electronically active and ultra-thin selective layer for OSN applications.

#### 2.2.3. Selective Layer Prepared by Sustainable Sources 

For preparing green TFC membranes, many researchers have tried to alternate the traditional petroleum-based monomers to the diverse sustainable sources. Figure 8 shows the promising candidates of the sustainable sources for preparing the eco-friendly selective layer, which are extracted from natural materials such as mussels, red onion, guava, starch, chitin shells, corns, vanillin bean, sumac leaves, etc. 

Among them, tannic acid [76,77,78,79,80], quercetin [81], morin hydrate [82], cyclodextrin [83,84,85], chitosan [86], and vanillin [87] played a well role as a reactive material, resulting in the formation of the stable selective layer having great performance. However, most of approaches have changed only one monomer. For example, cyclodextrins with different chemical structures were reacted with the petroleum-based chlorides including TMC or terephthaloyl chloride (TPC) [83]. In addition, they have still used toxic organic solvents such as *n*-hexane and toluene. 

To achieve the real green TFC membrane with solving the drawbacks, Park et al. recently reported that the eco-friendly selective layer with 30 nm thickness was prepared with the new green monomer combination of tannic acid and Priamine using less toxic organic solvent of *p*-cymene instead of *n*-hexane as shown in Figure 9 [76]. The reaction between tannic acid and Priamine lead to the formation of C=N and C-N covalent bonds via Schiff base-based and Michael reaction-based reactions, respectively. Additionally, the recycled PET porous membrane was used as a support membrane. Thus, the eco-friendly selective layer, recycled support, layer and less toxic solvents are enough to meet the requirements of the next-generation green TFC membrane. Moreover, it is the most promising candidate to replace the traditional petroleum-based selective layer prepared by MPD and TMC due to its rapid reaction time at room temperature. Priamine having long aliphatic chains endowed high contact angle values around 100°, leading to the outstanding permeance in non-polar solvent. 

Membrane materials for OSN TFC membranes and their performance against non-polar solvents including toluene, n-hexane, and n-heptane are summarized in Table 2. The TFC membranes are usually prepared using different methods, solvent systems, diverse monomers, and some additives like a nanomaterial including zeolites, silicalites, quantum dots (QD), and graphene oxides (GO). Performance test is conducted under a cross-flow or dead-end configuration system with diverse parameters such as pressure, temperature, agitation rate, and different concentration of solutes in various solvents [88]. Performance in the RO system is easy to compare each other since the fixed concentration of sodium chloride in water and operation conditions have been used [88]. However, it is very difficult to fairly compare membrane performance owing to more complex testing system with diverse solvent–solute combinations and operating conditions. Therefore, the standard protocol for performance testing is required.

## 3. Conclusions 

In this review paper, recently reported OSN TFC membranes are described to understand materials and research trends for the selective layer and porous support layer of TFC membranes. The porous support layer should be stable in the diverse organic solvent system and endured from the harsh operation condition more than 10 bar. In terms of performance, the proper choice of monomers and/or polymers as well as additives is the key point to fabricate the selective layer including much more interconnected voids, the enhanced microporosity, and hydrophilic/hydrophobic natures on highly porous support layer, resulting in the formation of the high permeable OSN TFC membranes with the proper rejection level. As mentioned above, the contorted monomers such as TTSBI and PIM are one of great candidate to provide relatively low chain packing and rigid polymer backbone, resulting in high free volume and microporosity [48,65]. In addition, Priamine including highly reactive amine groups and long aliphatic chains is a good example as a monomer to produce the hydrophobic membrane surface [78]. Furthermore, these could be also good ideas to obtain the hydrophobic surface by the use of fluoro-based materials as an additive [92,96] or a coating source [97] and the surface modification with hydrophobic materials [98]. 

Representative commercial OSN TFC membranes for non-polar solvents are composed of silicon polymer (polydimethylsiloxane, PDMS) selective layer on top of a cross-linked porous support [99,100]. Thickness of PDMS selective layer prepared by a coating method is about 1~2 μm [100], leading to the low permeance. On the other hand, the selective layer thickness of TFC membranes could be controlled less than 10 nm, which is beneficial for obtaining high-permeable membrane. We found that only a few OSN TFC membranes can be used in the separation and purification process of the non-polar solvent environments. Non-polar solvents such as toluene, *n*-hexane, and *n*-heptane have been widely used in crucial industries in our future, e.g., pharmaceutical and bio-industrial processes as well as fine chemical reaction processes. To give the huge impact on those kinds of applications and to be a game changer in green and cost-effective separation process, it is necessary to take concerns and to develop the next-generation OSN membranes, which can be used in the non-polar solvent environments. In addition, eco-friendly materials and green process with less toxic chemical are going to be keyword for designing the next-generation OSN TFC membranes since the environmental pollution problem become a big issue in industrial materials and processes.

## Figures and Tables

**Figure 1 membranes-11-00184-f001:**
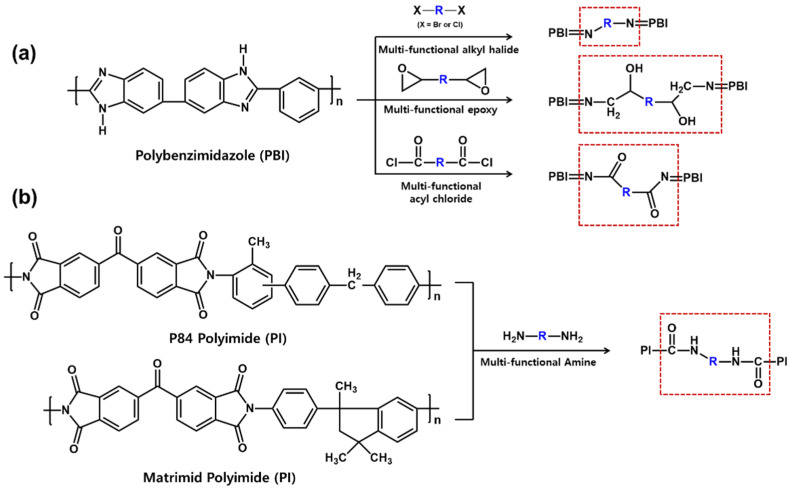
Cross-linking reaction scheme of (**a**) polybenzimidazole (PBI) and (**b**) polyimides (PI).

**Figure 2 membranes-11-00184-f002:**
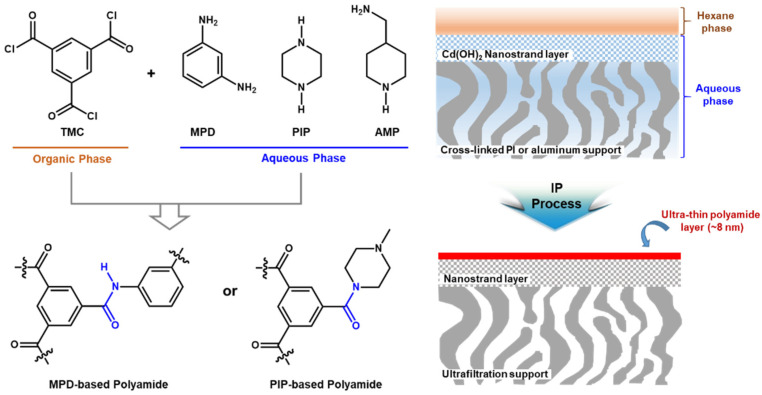
Ultra-thin *m*-phenylenediamine (MPD)-based and/or piperazine (PIP)-based polyamide selective layer with the ultrafast solvent transport.

**Figure 3 membranes-11-00184-f003:**
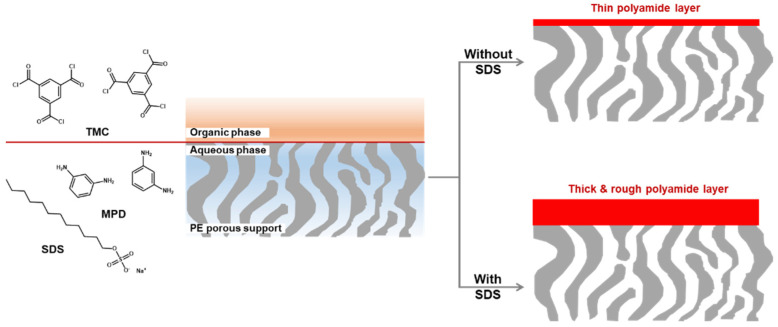
MPD-based polyamide selective layer on top of polyethylene (PE) porous support membrane.

**Figure 4 membranes-11-00184-f004:**
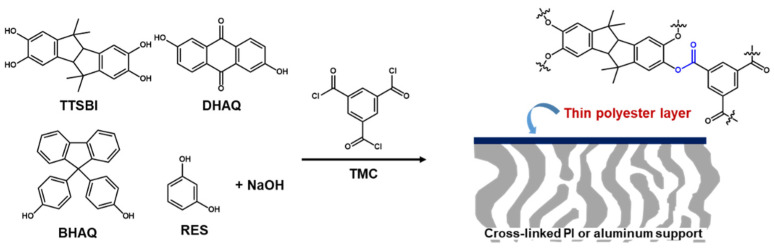
Phenol-based polyester selective layer with interconnected voids and the microporosity.

**Figure 5 membranes-11-00184-f005:**
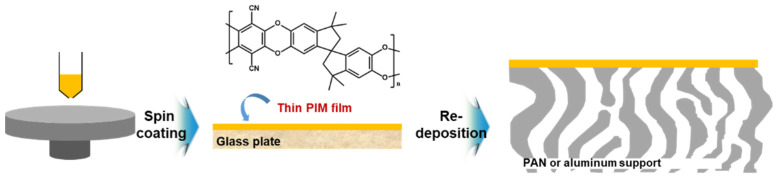
PIM-based ultra-thin films with high free volume donated from its interconnected intermolecular voids.

**Figure 6 membranes-11-00184-f006:**
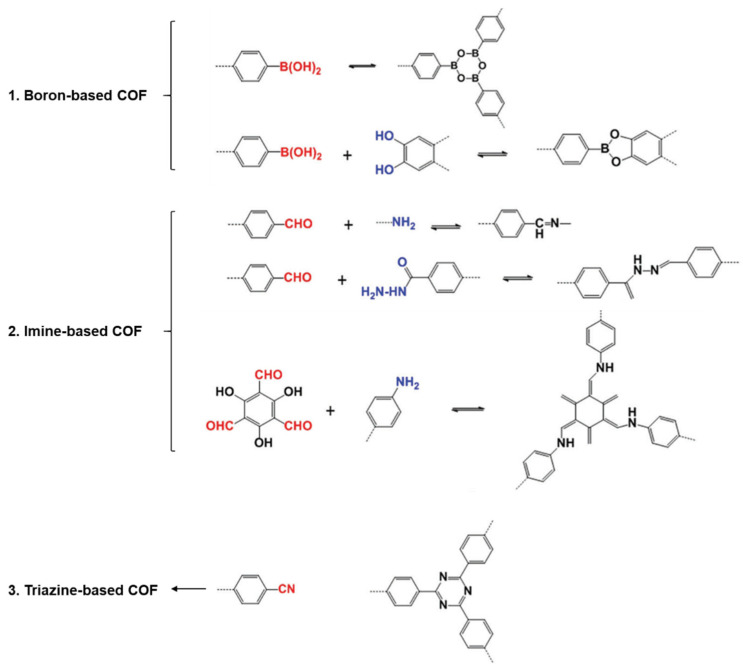
Chemical structures of diverse types of covalent organic frameworks (COFs).

**Figure 7 membranes-11-00184-f007:**
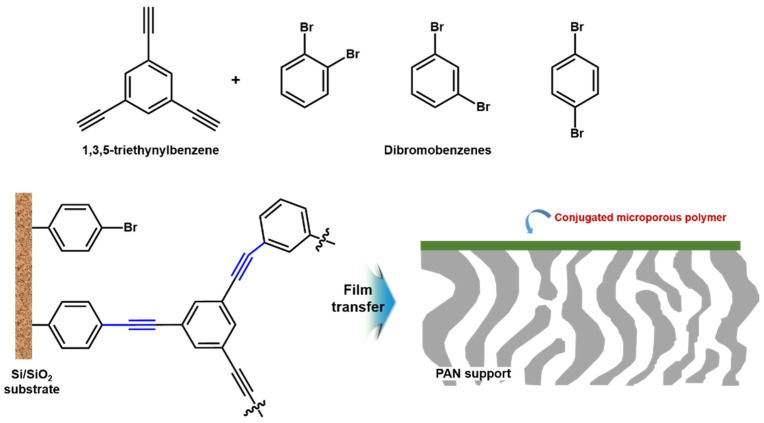
Conjugated microporous polymers (CMP)-based ultra-thin film with microporous voids and conjugated rigid-backbone structures.

**Figure 8 membranes-11-00184-f008:**
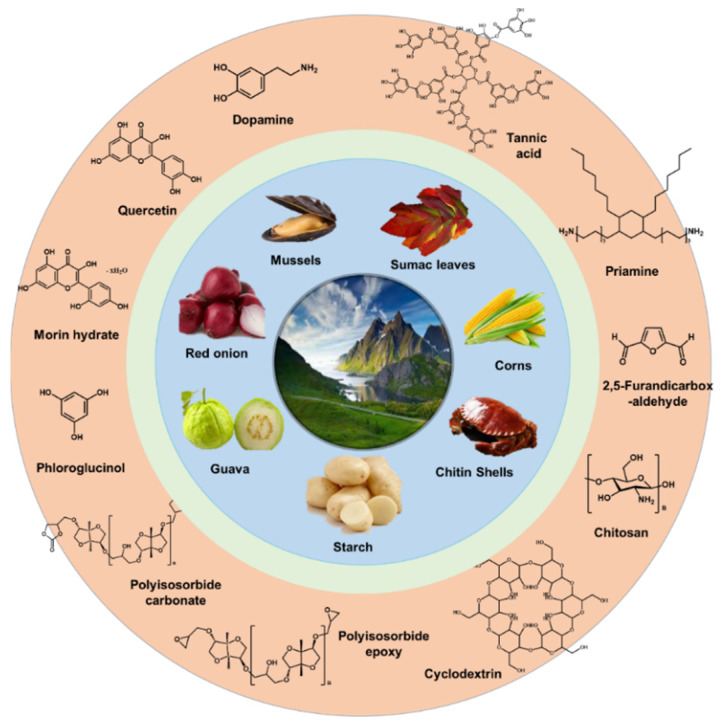
The candidates of the sustainable sources for preparing green selective layer of thin-film composite (TFC) membranes.

**Figure 9 membranes-11-00184-f009:**
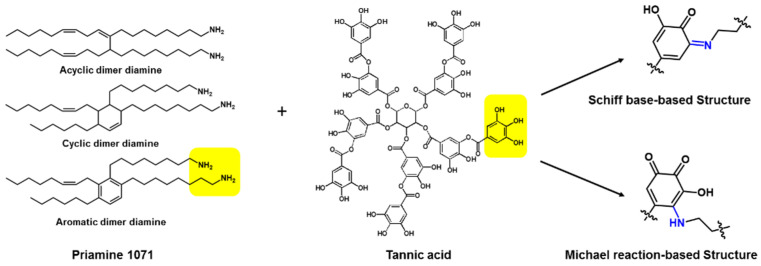
Green source-based ultra-thin film with hydrophobic natures.

**Table 1 membranes-11-00184-t001:** Chemical names and structures of common support membranes.

Name	Structure	Reference
Polysulfone(PSF)	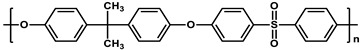	[14]
Polyethersulfone(PES)	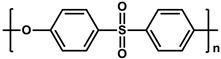	[20]
Poly(ether ether ketone)(PEEK)	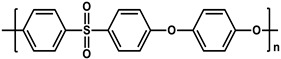	[22]
Polyacrylonitrile(PAN)	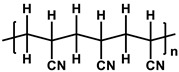	[25]
Poly(vinylidene fluoride)(PVDF)	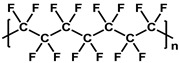	[27]
Polyethylene(PE)	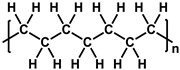	[30]
Polypropylene(PP)	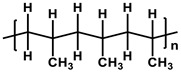	[31]
Polyethylene terephthalate(PET)	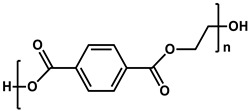	[34]
Cellulose	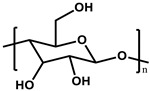	[35]

**Table 2 membranes-11-00184-t002:** Membrane materials and performance in non-polar solvents such as toluene, *n*-hexane, and *n*-heptane.

Membrane No.	Fabrication			Performance					Reference
	Materials (Selective Layer/ Support)	Method	Solvent System	Solvent	Permeance (L m^−2^ h^−1^ bar^−1^)	Solute	Solute MW (g mol^−1^)	Rejection (%)	
1	Nanoparticle/PI	Coating	Methanol	Toluene	0.6	Styrene dimers	220	90	[89]
2	Zeolite-filled PDMS/PAN	Coating	Hexane/water		0.58	Wilkinson catalyst	925	>97	[90]
3	Silicalite-filled PDMS/PI	Coating	Hexane		0.9	Bromothymol blue	624	80	[91]
4	Fluoro-functional PA/PEEK	IP ^(1)^	Hexane/water		2.0	Styrene dimers	236	98	[92]
5	PIM/PAN	Dip coating	Chloroform		7.1	Polystyrene	800	90	[63]
6	QD-based PA/PAN	IP	Hexane/water		2.5	Acid yellow 14	450	90	[93]
7	TA-based polyimine/PET	IP	*p*-cymene/water		3.5	Styrene dimers	235	75	[76]
8	CMP/PAN	Grafting	Toluene/triethylamine	*n*-Hexane	31.7	Protoporphyrin IX	562	90	[75]
9	GO-filled PA	IP	Hexane/water		0.1	Rhodamine B	475	95	[94]
10	QD-based PA/PAN	IP	Hexane/water		51	AY79 Dyes	1280	99	[93]
11	PIM/PAN	Spin coating	Chloroform	*n*-Heptane	18	Hexaphenylbenzene	535	86~90	[8]
12	Ti_3_C_2_T_x_-filled PA/PAN	IP	Hexane/water		1.8	PEG	200	92	[95]
13	PIM/PAN	Dip coating	Chloroform		2.5	polystyrene	900	90	[63]
14	TA-based polyimine/PET	IP process	*p*-cymene/water		2.5	Styrene dimers	235	91	[76]

^(1)^ Interfacial polymerization.

## Data Availability

Not applicable.

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
