# Peer review of "Thin-Film Composite Nanofiltration Membranes for Non-Polar Solvents"

_membranes, 2021, doi:10.3390/membranes11030184_

Round 1
Reviewer 1 Report
The review by Lee and colleagues gives a detailed insight into non-polar solvent nanofiltration using TFC membranes. This field has a growing interest over the last years. Therefore a review on this topic is timely, and it is of interest to the readers of the journal. In general the review is well-written and comprehensive but can be improved by addressing the following minor comments.
1, The original comparison tables and figures are very useful. The authors could complement this by a final comparison table that allows the quick review of permeances and rejections of the different hydrophobic membranes, their fabrication methods, monomers, solvent systems etc.
2, Superhydrophobic membrane surfaces made from naturally derived hydrophobic materials should also be briefly mentioned in the review (ACS-SCE, 2017, 5, 11362).
3, Some recent literature on hydrophobic fluorine containing literature is missing from the review (ACS-APM, 2019, 1, 472; JMS 2021, 119112).
4, The guideline for testing OSN membranes could be briefly mentioned in the review (Green Chem., 2020, 22, 3397).
5, The field of non-polar solvent nanofiltration is fairly scarce, and therefore the alternatives to TFC approach should also be briefly mentioned, for instance hydrophobic PIMs (Science, 2020, 369, 310).
6, There is more work on green TFC for hydrophobic OSN that could also be mentioned in the review for completeness (Sep Purif Technol, 2021, 263, 118394).
7, In lines 115-130, the authors mention about the different crosslinking of PBI and PI for making them stable for TFC-OSN applications. Besides the mentioned direct crosslinking, other methods should also be mentioned, such as highly stable support can also be achieved via IPNs and TRPs (ACS Nano 2019, 13, 125; Chem Eng J, 2021, 409, 128206; JMS 2018, 550, 322).
8, There are recent efforts to integrate macrocycles for precise molecular sieving in the nanofiltration range, and some are hydrophobic TFCs, which could be briefly discussed (Desalination, 2021, 500, 114861).
Author Response
Dear Reviewer,
Thank you for kind review comments.
Please find the response for each comment in the attached file.
best regards,
Sang-Hee Park.

Reviewer 2 Report
The authors review the application of TFC NF membranes in solvent recycle & exchange, concentrations and purification of chemicals and pharmaceuticals in organic solvent environments. The review is well written and carries fundamental information that will add value to Membranes. It outlines major research work that has been reported with regards to the use of organic solvent nanofiltration (OSN) thin film composite (TFC) membranes for non-polar solvents. Having said that, the following is suggested to improve the quality of the review:
Page 1, line 21: please delete “w” in the abstract before the paragraph with keywords.
Page 3, line 96: please avoid the use of the term “significant” unless statistical results are provided.
Page 6, line 207 – 209: what is “thick thickness”?
In Figure 8 the authors presented candidates of the sustainable sources for preparing green selective layer of TFC membranes. Please explain what the future of these materials is in making a break-through in industrial application compared to other additives.
Finally, it would be great if the authors would add few examples of real-life industrial application of these TFC membranes for non-solvents, indicating which polymers are mainly used and provide explanations why other polymers are not preferred.

Author Response

(The authors gave the same response as above.)

Reviewer 3 Report
According to the authors of the manuscript S. Lee et al "Thin Film Composite Nanofiltration Membranes for Non-polar Solvents" the work focuses on the most recent advances in the field of obtaining membranes with a thin selective layer. The work is no longer an analysis but a listing of options for obtaining films. In my opinion, the review article should contain a deeper analysis of the considered results. Thus, the manuscript omits the question of the mechanical characteristics of the membranes described. Structure and morphology are considered superficially. The conclusions of the manuscript need to be drastically redone.
Line 21. Remove "w".
Lines 99-102. It makes sense to add cellulose to this list.
Line 113. Delete Table 1.
Lines 207, 208. Rephrase this "thick thickness".
Line 323, 325. In the case of chitosan, is it also a "monomer"?
Author Response

(The authors gave the same response as above.)

Round 2
Reviewer 3 Report
The authors responded to the comments of the reviewer and made changes to the manuscript. The submitted manuscript can then be reviewed by the editor for publication.